# A Spectral Viewpoint on Continual Relation Extraction

**Huy Huu Nguyen**[1], **Chien Van Nguyen**[2], **Linh Ngo Van**[3*]
**Luu Anh Tuan**[4] and **Thien Huu Nguyen**[2]
[1] VinAI Research, Vietnam
[2] Department of Computer Science, University of Oregon, Eugene, OR, USA
[3] Hanoi University of Science and Technology, Hanoi, Vietnam
[4] Nanyang Technological University, Singapore
v.huynh16@vinai.io, {chienn,thien@cs}@uoregon.edu
linhnv@soict.hust.edu.vn, anhtuan.luu@ntu.edu.sg

## Abstract

Continual Relation Extraction (CRE) aims to continuously train a model to learn new relations while preserving its ability on previously learned relations. Similar to other continual learning problems, in CRE, models experience representation shift, where learned deep space changes in the continual learning process, which leads to the downgrade in the performance of the old tasks. In this work, we will provide an insight into this phenomenon under the spectral viewpoint. Our key argument is that, for each class shape, if its eigenvectors (or spectral components) do not change much, the shape is well-preserved. We then conduct a spectral experiment and show that, for the shape of each class, the eigenvectors with larger eigenvalue are more preserved after learning new tasks which means these vectors are good at keeping class shapes. Based on this analysis, we propose a simple yet effective class-wise regularization that improve the eigenvalues in the representation learning. We observe that our proposed regularization leads to an increase in the eigenvalues. Extensive experiments on two benchmark datasets, FewRel and TACRED, show the effectiveness of our proposed method with significant improvement in performance compared to the state-of-the-art models. Further analyses also verify our hypothesis that larger eigenvalues lead to better performance and vice versa.

## 1 Introduction

Relation Extraction (RE) problem is the basis of many NLP tasks, such as Question Answering (Sorokin and Gurevych, 2017), Knowledge Graph Construction (Luu et al., 2014, 2016; Baldini Soares et al., 2019), Definition Extraction (Veyseh et al., 2020) and event relation extraction (Man et al., 2022; Lai et al., 2022). In particular, a relation extraction system is expected to classify semantic relation between two entities mentioned in

the given context. To address this problem, several methods have been proposed and have achieved remarkable results (Nguyen and Grishman, 2015; Zhou et al., 2016; Pouran Ben Veyseh et al., 2020; Zheng et al., 2023). Nevertheless, most previous RE studies only considered the traditional setting where the set of relations is pre-defined and fixed during the training and testing phases. This setting is not practical as new relations of interest might emerge during deployment time of RE systems in practice, requiring the models to adapt their operations to accommodate new types.

Recently, Continual Relation Extraction (CRE) has attracted considerable attention in the literature (Wang et al., 2019; Han et al., 2020; Cui et al., 2021; Zhao et al., 2022), aiming to learn new relations from incoming data. CRE methods often suffer from Catastrophic Forgetting (CF) phenomenon where their performance on previous relations reduce significantly when learning new relations. One key issue that causes CF is the representation shift that occurs when models learn new knowledge. Existing works have made significant progress in mitigating this phenomenon (Phan et al., 2022; Van et al., 2022; Nam et al., 2023). Different from those works, we analyze the representation shift using spectral decomposition. We argue that the eigenvectors play an important role in capturing the shape of each class. As such, if the eigenvectors of a class shape adjust much after exposing new tasks, it means the class shape also changes significantly. We give an intuition for our argument in figure 1. Based on this reasoning, we conduct an experiment to see how eigenvectors change in the continual learning process. We find that the eigenvectors with larger eigenvalue are preserved well when learning new tasks. In other words, these eigenvectors are good at keeping class shapes and hence avoiding the representation shift.

Based on these experiments, we propose a class-wise regularization that aims to boost the eigen-

---

*Corresponding Author

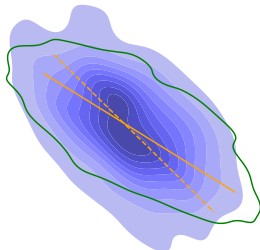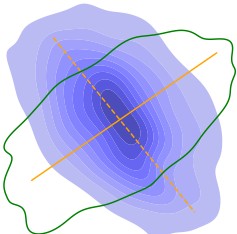

Figure 1: A demonstration for the impact of the adjustment of the first component vector. The class shape before updating is blue and the updated one is green. The first component before updating is the dashed line while the new one is the solid line. **Left**: The small adjustment keeps the shape unchanged. **Right**: The large adjustment leads to a bad representation shift.

values. This regularization is inspired by some recent works on dimensional collapse and feature decorrelation (Bardes et al., 2022), which try to make different dimensions of the embedding uncorrelated and avoid small eigenvalues effectively. To enhance understanding, we theoretically show why feature decorrelation helps increase the eigenvalues.

In summary, our work makes the following contributions:

- We conduct a thorough analysis and provide an insightful view of the representation space in continual relation extraction setting via the spectral perspective. We find that the spectral components with larger eigenvalues are less forgettable and useful in preserving class shapes.

- A simple yet effective class-wise feature decorrelation regularization is proposed with the goal of boosting the eigenvalues of the representation for each class. Furthermore, we theoretically demonstrate why feature decorrelation helps boost eigenvalues.

Our extensive experiments demonstrate that our model achieves new state-of-the-art performance on two CRE benchmarks, FewRel and TACRED. We also conduct further analysis to verify the efficacy of our idea in boosting the eigenvalues.

## 2 Problem Formulation and Base Model

**Problem Formulation**: In continual relation extraction, models are trained on a sequence of tasks $\{T_1, T_2, ..., T_k\}$, where the *k-th* task is one traditional relation extraction task and has its own train-

ing set $D_k$ and relation set $R_k$. Formally, each task $T_k$ is a supervised classification task with training set $D_k$ containing N samples $\{(x_i, y_i)\}_{i=1}^N$, where $x_i$ is the input sentence and entity pair, and $y_i$ is the relation label. As a general goal of continual learning, a CRE system is expected to perform well on both the current task and all previous tasks. Hence, models are required to classify each relation into a known relation set $\tilde{R}_k$, where $\tilde{R}_k = \bigcup_{i=1}^k R_i$.

We adapt an episodic memory module to store a few examples of historical tasks. Each relation has its own memory module, *i.e.*, a memory module for relation $r$ is a set $M_r = \{(x_i, y_i)\}_{i=1}^O$ storing O samples that are representatives for relation $r$, where O is the pre-defined number of samples to be stored.

**Base Model and Objective Function**: For modelling details, we use one pretrained language model, *i.e.*, BERT (Devlin et al., 2018) as a feature extractor and a learnable linear classifier on top of the feature extractor. Following previous work (Han et al., 2020; Cui et al., 2021), we use the CE loss $L_{CE}$ as the objective function for training task $T_k$:

$$L_{CE} = \sum_{i=1}^{|D_k|} -\log P(y_i|x_i),$$

where $(x_i, y_i) \in D_k$

## 3 Spectral Analysis on The Representation Space

### 3.1 Spectral Analysis Setups

Inspired by (Zhu et al., 2021), in this section we aim to explore the changes in class shapes before and after learning new tasks. Concretely

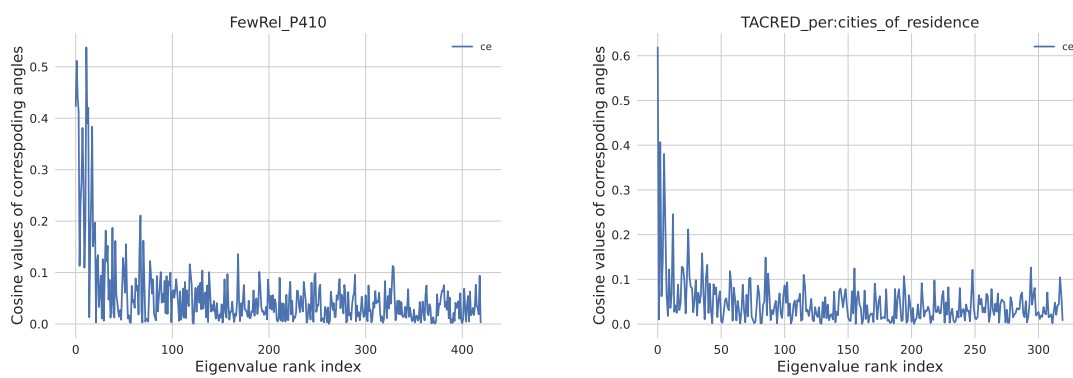

Figure 2: Absolute cosine values of corresponding angles of two classes *P410* and *per:cities of residence*, which we choose randomly from the first task of FewRel and TACRED, respectively.

| FewRel | | | | | | | | | | |
|---|---|---|---|---|---|---|---|---|---|---|
| Model | T1 | T2 | T3 | T4 | T5 | T6 | T7 | T8 | T9 | T10 |
| **EA-EMR** | 89.0 | 69.0 | 59.1 | 54.2 | 47.8 | 46.1 | 43.1 | 40.7 | 38.6 | 35.2 |
| **CML** | 91.2 | 74.8 | 68.2 | 58.2 | 53.7 | 50.4 | 47.8 | 44.4 | 43.1 | 39.7 |
| **EMAR + BERT** | 98.1 | 94.3 | 92.3 | 90.5 | 89.7 | 88.5 | 87.2 | 86.1 | 84.8 | 83.6 |
| **RP-CRE** | 97.8 | 94.7 | 92.1 | 90.3 | 89.4 | 88.0 | 87.1 | 85.8 | 84.4 | 82.8 |
| **CRL** | 98.1 | 94.6 | 92.5 | 90.5 | 89.4 | 87.9 | 86.9 | 85.6 | 84.5 | 83.1 |
| *EMAR + ACA* | **98.3** | **95.0** | 92.6 | 91.3 | 90.4 | 89.2 | 87.6 | **87.0** | **86.3** | **84.7** |
| **Ours** | **98.3** | 94.7 | **93.1** | **91.4** | **90.6** | **89.4** | **87.9** | 86.9 | 85.4 | 84.3 |
| TACRED | | | | | | | | | | |
| Model | T1 | T2 | T3 | T4 | T5 | T6 | T7 | T8 | T9 | T10 |
| **EA-EMR** | 47.5 | 40.1 | 38.3 | 29.9 | 24 | 27.3 | 26.9 | 25.8 | 22.9 | 19.8 |
| **CML** | 57.2 | 51.4 | 41.3 | 39.3 | 35.9 | 28.9 | 27.3 | 26.9 | 24.8 | 23.4 |
| **EMAR + BERT** | **98.3** | 92.0 | 87.4 | 84.1 | 82.1 | 80.6 | 78.3 | 76.6 | 76.8 | 76.1 |
| **RP-CRE** | 97.5 | 92.2 | 89.1 | 84.2 | 81.7 | 81.0 | 78.1 | 76.1 | 75.0 | 75.3 |
| **CRL** | 97.7 | 93.2 | 89.8 | 84.7 | 84.1 | 81.3 | 80.2 | 79.1 | 79.0 | 78.0 |
| *EMAR + ACA* | 98.0 | 92.1 | **90.6** | 85.5 | 84.4 | 82.2 | 80.0 | 78.6 | 78.8 | 78.1 |
| **Ours** | 98.1 | **93.8** | 89.8 | **85.8** | **84.4** | **83.4** | **81.6** | **79.9** | **79.7** | **79.1** |

Table 1: Accuracy (%) on all observed relations (which will continue to accumulate over time) at the stage of learning current task. Other results are directly taken from (Wang et al., 2022). We show the best results in **boldface** and the second best ones in underlines (Note that all models do not use class augmentation except **EMAR + ACA**)

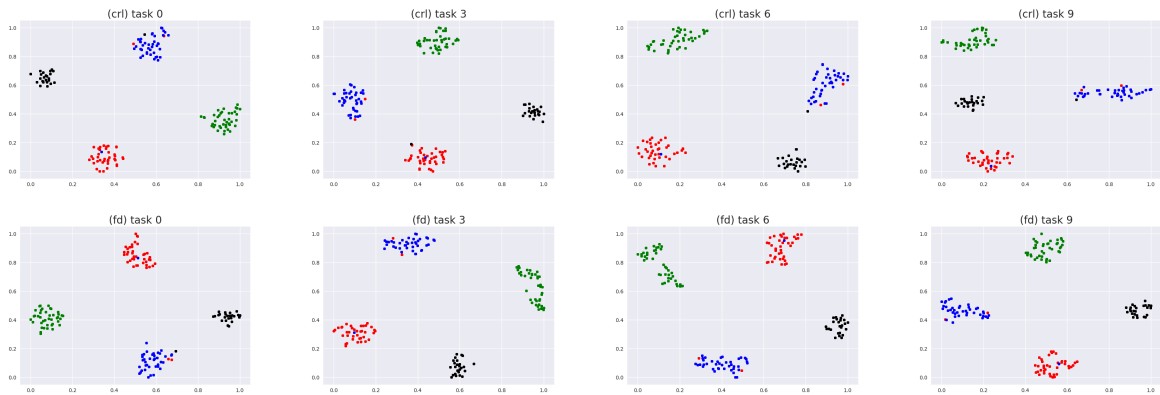

Figure 3: A visualization of relation representation learned from task 1 of the test set by CRL (**Top**) and our model (**Bottom**) at different task

speaking, we monitor how every eigenvector of deep feature space is adjusted after updating new knowledge thanks to spectral decomposition. Formally, we first train a feature extractor on dataset $\mathcal{D}_{old} = \{(x_i, y_i)\}_{i=1}^{n}$ of current task, denoted as $\mathcal{F}_{old}$. After finetuning $\mathcal{F}_{old}$ on new dataset $\mathcal{D}_{new} = \{(x_i, y_i)\}_{i=1}^{n}$, we obtain an updated extractor, denoted as $\mathcal{F}_{new}$. To measure the sensi-

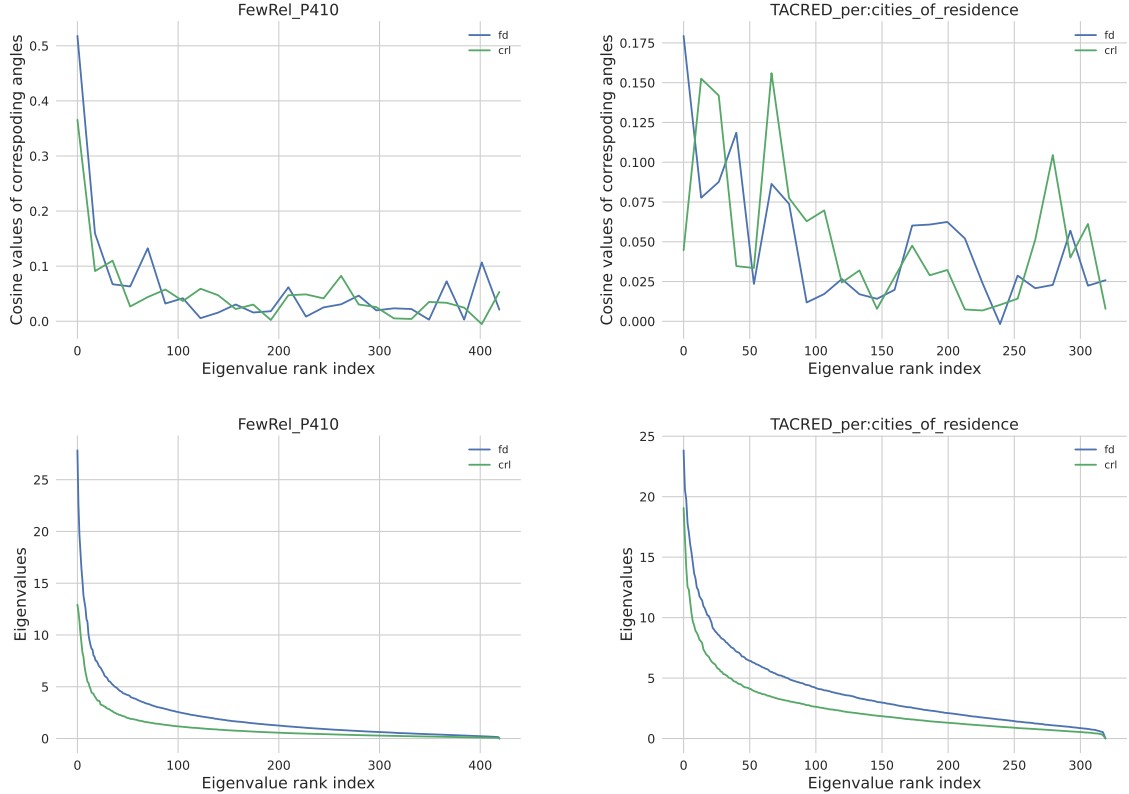

Figure 4: **Top**: Absolute cosine values of corresponding angles of two classes *P410* and *per:cities of residence* between our model and the SOTA. **Bottom**: Distribution of eigenvalues of two classes *P410* and *per:cities of residence* for our model and the SOTA one . (Lines in cosine figures are smoothen for a better view)

tivity of learned class shapes, we use old dataset $\mathcal{D}_{old}$ and map it into old space and new space using $\mathcal{F}_{old}$ and $\mathcal{F}_{new}$ respectively. Let $\mathcal{F}_{old}(x_i)$ and $\mathcal{F}_{new}(x_i)$ be mapped features of a sample $x_i$ on old and new spaces, we do spectral decomposition on the correlation matrix as follows:

$$\frac{1}{n}\sum_{i=1}^{n}\tilde{\mathcal{F}}(x_i)\tilde{\mathcal{F}}(x_i)^T = \sum_{j=1}^{d}\mathbf{u}_j\lambda_j\mathbf{u}_j^T,$$

where in the L.H.S, $\tilde{\mathcal{F}}$ is the mapped feature $\mathcal{F}$ after normalizing, while in the R.H.S the $\lambda_j$ is the *j-th* eigenvalue in the sorted list of $d$ eigenvalues in descending order and $u_j$ is its corresponding eigenvector. Applying this decomposition on both old representations and new ones, $\mathcal{F}_{old}(x_i)$ and $\mathcal{F}_{new}(x_i)$, we obtain two set of eigenvectors $\{\mathbf{u}_{old,1},...,\mathbf{u}_{old,d}\}$ and $\{\mathbf{u}_{new,1},...,\mathbf{u}_{new,d}\}$.

In order to investigate the change of the eigenvectors, Zhu et al. (2021) introduced the definition of **Corresponding Angle**: given two groups of eigenvectors, $\{\mathbf{u}_{old,1},...,\mathbf{u}_{old,d}\}$ and $\{\mathbf{u}_{new,1},...,\mathbf{u}_{new,d}\}$, **corresponding angle** presents the angle between two eigenvectors corre-

sponding to the same eigenvalue index. The cosine value of the corresponding angle is computed as follows:

$$cos(\psi_j) \overset{(1)}{=} \frac{\langle\mathbf{u}_{old,j},\mathbf{u}_{new,j}\rangle}{\|\mathbf{u}_{old,j}\|\cdot\|\mathbf{u}_{new,j}\|} \overset{(2)}{=} \langle\mathbf{u}_{old,j},\mathbf{u}_{new,j}\rangle$$

where $\mathbf{u}_{old,j}$ is the *j-th* eigenvectors with the *j-th* largest eigenvalue in the old feature space, and similarly for $\mathbf{u}_{new,j}$. Additionally, because $\|\mathbf{u}_{old,j}\| = \|\mathbf{u}_{new,j}\| = 1$, the $\overset{(2)}{=}$ is trivial.

## 3.2 Changes in eigenvectors

We conduct an experiment to explore which part, *i.e.*, which direction of the representation space carries helpful knowledge in CRE setting. We leverage the first two tasks of the FewRel and TACRED datasets to examine the feature space. Figure 2 shows the absolute cosine values of the corresponding angles between the old and new eigenvectors. It is clear that, after updating, eigenvectors with large eigenvalues adjust insignificantly and are therefore good at preserving the shape of the data distribution, while ones with small eigenvalues tend to

move towards some different direction and can be considered as noisy and forgettable directions.

### 3.3 Class-wise Decorrelation Feature Regularization

With the previous finding, our goal is to boost the eigenvalues of the distribution shape of each class. Hua et al. (2021) indicated the relationship between strong correlation and dimensional collapse, which leads to the idea of feature decorrelation. Based on this finding, we propose a simple technique that enlarges eigenvalues. Formally, in each batch, for a given label $r$, the correlation matrix of $r$ is estimated as:

$$K^{(r)} = \frac{1}{n-1} \sum_{i=1}^{n} (Z_i^{(r)} - \bar{Z}^r)(Z_i^{(r)} - \bar{Z}^r)^T,$$

where $Z_i^{(r)}$ is the encoder output representation on the *i-th* data point of class $r$ while the mean vector of $r$ is denoted as $\bar{Z}^r = \frac{1}{n} \sum_{i=1}^{n}(Z_i^{(r)})$. The data representations is explicitly constrained by a class-wise feature decorrelation regularization:

$$L_{FD} = \sum_{r \in R_k} \sum_{i \# j} K_{i,j}^{(r)2}$$

We give the theoretical proof on how feature decorrelation enlarges the eigenvalues in Appendix B.1. The overall objective function is given by:

$$L_{overall} = L_{CE} + \mu L_{FD},$$

where $\mu$ are the hyper-parameters controlling the importance of each term in the overall loss.

## 4 Experiments

### 4.1 Experiments Setups

**Datasets**: Similar to (Zhao et al., 2022), we perform our experiments on two standard English benchmark datasets: **FewRel** and **TACRED**. More details about two datasets is given at A.1

**Implementation Details**: Following (Cui et al., 2021), relations are randomly divided into 10 clusters to simulate 10 tasks. We use average accuracy on all seen tasks to measure model performance on the CRE task as previous work (Han et al., 2020). In terms of environment and configuration, we train all models on the same task sequence used in (Cui et al., 2021; Zhao et al., 2022) by setting exactly the same random seed to make a fair comparison. For the convenience of reproduction we provide the details of hyper-parameters settings at A.3.

### 4.2 Experimental Results

**Base results**: We report the results of our proposed method and some baselines in Table 1. We give the details of the baselines in A.2. It can be seen from the table that, compare with other models which do not use class augmentation ACA (Wang et al., 2022), our model sets new SOTA on two CRE benchmark datasets. Our model performs better than the current SOTA, CRL by 1.2 % on FewRel and 1.1% on TACRED. Our model without ACA performs on par with the EMAR + ACA on **FewRel** and still outperforms it remarkably on **TACRED**.

**Effectiveness of the regularizer on eigenvalues**: We run the same spectral analysis on both our model and see that it produces larger eigenvalues compared to the SOTA model, CRL. The reason we do not conduct experiments with ACA because they use synthesis classes that leads to different behaviours in deep space compared to normal methods. We display changes in class shapes using t-SNE in figure 3 and also provide figures about the corresponding angles in figure 4 to verify the effectiveness of our regularizer in keeping class shapes. In detail, in figure 3, we show the representation of the test set data from four classes of the first task of **TACRED** after learning task 0, 3, 6, 9. Our discussion is that: in a long run, the class shapes in CRL tend to be thinner and longer while ones trained with our model do not suffer it. The gap in eigenvalues (Figure 4) shows that our feature decorrelation regularization actually increases the eigenvalues and the cosine value of corresponding angles.

## 5 Conclusion

In this work, we conduct a thorough study on the change of distribution shapes in the CRE problem through spectral analysis and observe that eigenvectors with larger eigenvalues are less forgettable. Based on these findings, we introduce a class-wise feature decorrelation regularization with the goal of boosting eigenvalues. Our theoretical analysis shows the effectiveness of our proposed method in handling low eigenvalues. Furthermore, our extensive experiments on two benchmark datasets show the superior performance of our method. In the future, we will extend our analysis to other Information Extraction tasks, such as Entity Mention Detection (Nguyen et al., 2016) and Event Classification (Lai et al., 2020; Hao et al., 2023).

## Limitations

- Although we do an analysis to indicate that eigenvectors with larger eigenvalues carry more helpful features, there is a lack of interpretation in these directions. It would be preferable to provide some examples to demonstrate the superiority of these directions and make them more understandable.

- Like all prior works about CRE, we only focus on classifying a pair of entities into relation types when given those entities in a context. To aim for a complete solution for CRE, the problem of named entity recognition should be studied in the continual learning scenarios.

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

## A  Experiments Details

### A.1  Datasets

We perform our experiments on two English benchmark datasets: **FewRel** and **TACRED** The training-test-validation split ratio is 3:1:1. **FewRel** (Han et al., 2018) is a RE dataset that contains 80 relations and 56,000 samples in total. To make it a continual RE dataset, we follow the settings in (Wang et al., 2019) and use the original train set and validation set for our experiments. **TACRED** (Zhang et al., 2017) is an imbalanced RE dataset that contains 42 relations (including $no\_relation$) and 106,264 samples. Following the experiment settings by (Cui et al., 2021), to reduce the imbalance, we remove the $no\_relation$ class and limit the number of training samples of each relation to 320 and the number of test samples of each relation to 40.

### A.2  Baselines

We compare our proposed methods with several state-of-the-art CRE baselines:

- **EA-EMR** (Wang et al., 2019) proposed a memory replay and embedding alignment technique to mitigate catastrophic forgetting.

- **EMAR** (Han et al., 2020) introduced a memory activation and reconsolidation to preserve old knowledge.

- **CML** (Wu et al., 2021) proposed a curriculum-meta learning method to handle the order-sensitivity and catastrophic forgetting in CRE.

- **RP-CRE** (Cui et al., 2021) learns a memory network with the goal of refining sample embeddings with relation prototypes, thereby avoid catastrophic forgetting.

- **CRL** (Zhao et al., 2022) maintains learned knowledge by introducing contrastive replay mechanism and knowledge distillation.

- **ACA** (Wang et al., 2022) proposed an adversarial class augmentation mechanism to make learned models more robust representations.

### A.3  Reproducibility Checklist

- **Source code with specification of all dependencies, including external libraries**: Our source code with necessary documentation for reproducibility will be released upon acceptance of the paper.

- **Description of computing infrastructure used**: In this work, we use a single Tesla A100 GPU with 100GB memory operated by Ubuntu Server 18.04.3 LTS for all experiments. PyTorch 2.0 and Huggingface-Transformer 4.29.2 (Apache License 2.0) (Wolf et al., 2019) are used to implement the models.

- **Average runtime**: Training each round on average takes 88 minutes for FewRel dataset and 18 minutes for TACRED dataset. For each task, in both Initial training for new task phase and Memory replay phase we train the model for 10 epochs following the experiments by (Zhao et al., 2022)

- **Number of parameters in the model**: There are approximately 110 million parameters in total, including 110 million from the feature extractor and very few parameters, compared to the feature extractor, from the softmax classifier.

- **Bounds for each hyper-parameter**: To tune the proposed objective function, we choose $\mu$ from $[0.05, 0.1, 0.2, 0.25, 0.5, 1.0, 1.25]$. All the hyper-parameters are selected based on F1 scores on the validation set.

- **The method of choosing hyper-parameter values and the criterion used to select among them**: We choose the hyper-parameters for the proposed model using manual tuning.

- **Hyperparameter configurations for best-performing models**: In our model, we use the following values for the hyper-parameters: learning rate $1e$-5 for the encoder and $1e$-3 for the classifier with the AdamW optimizer; 32 for the mini-batch size; $\mu = 0.05$ for FewRel and $\mu = 0.1$ for TACRED. The memory size of each task is 10.

## B  The Feature Decorrelation Regularizer

### B.1  Theoretical proof for the effect of our loss on eigenvalues

Without the loss of generality, we give a proof for our loss $L_{FD}$ on a specific class $r$, denoted as $L_{FD}^{(r)}$:

$$L_{FD}^{(r)} = \sum_{i \# j} K_{i,j}^{(r)^2}$$

Suppose that the correlation matrix of $r$: $K^{(r)} \in R^{d \times d}$, defined in subsection 3.3, have $\{\lambda_i\}_{i=1}^{d}$ as its eigenvalues, we have:

$$\sum_{i=1}^{d} \lambda_i = trace(K^{(r)}) = d \qquad (1)$$

Above equalities hold because the entries on the main diagonal of a correlation matrix are 1's. Now, consider our proposed loss $L_{FD}^{(r)}$ , we have:

$$
\begin{aligned}
L_{FD}^{(r)} &= \sum_{i \# j} K_{i,j}^{(r)^2} \\
&= \|K^{(r)} - diag(K^{(r)})\|_F^2 \\
&= \|K^{(r)} - I_d\|_F^2
\end{aligned}
\qquad (2)
$$

Recall that the Frobenius norm is equal to the sum of its squared eigenvalues. The problem now is to find out which is the eigenvalue of the matrix $K^{(r)} - I_d$. In fact, its eigenvalues are $\{\lambda_i - 1\}_{i=1}^{d}$. Therefore, we have:

$$\|K^{(r)} - I_d\|_F^2 = \sum_{i=1}^{d} (\lambda_i - 1)^2 \qquad (3)$$

Moreover, by equation 1, we rewrite above equation as:

$$\|K^{(r)} - I_d\|_F^2 = \sum_{i=1}^{d} (\lambda_i - \frac{1}{d} \sum_{j=1}^{d} \lambda_j)^2 \quad (4)$$

Next, plug equation 4 back to equation 2, we have:

$$
\begin{aligned}
L_{FD}^{(r)} &= \sum_{i \# j} K_{i,j}^{(r)^2} \\
&= \sum_{i=1}^{d} (\lambda_i - \frac{1}{d} \sum_{j=1}^{d} \lambda_j)^2
\end{aligned}
\qquad (5)
$$

Note that our regularizer increases the number of eigenvectors with large eigenvalues by penalizing large eigenvalues. In fact, it reduces top eigenvalues and boosts the eigenvalues with lower indices.