# OpenReview forum: "A Spectral Viewpoint on Continual Relation Extraction"
_EMNLP/2023/Conference — EMNLP 2023 Findings_

### Official Review · Reviewer_rW8k · 2023-08-02

**Soundness:** 3

**Excitement:**

4: Strong: This paper deepens the understanding of some phenomenon or lowers the barriers to an existing research direction.

**Paper Topic And Main Contributions:**

The paper proposes a spectral viewpoint on continual relation extraction (CRE). The paper argues that the eigenvectors of the deep feature space play an important role in preserving the shape of each class and that eigenvectors with larger eigenvalues are less forgettable and useful in avoiding representation shift. Based on this point, the paper introduces a class-wise feature decorrelation regularization that boosts the eigenvalues of the representation for each class. The paper also provides theoretical and empirical evidence for the effectiveness of the proposed method, which achieves reasonable improvement on two benchmark datasets.

**Questions For The Authors:**

1.	The paper does not provide much interpretation or intuition for the eigenvectors with larger eigenvalues, and what kind of features they capture. It would be preferable to provide some examples or visualizations from a high-level viewpoint.
2.	The paper does not compare or discuss the relation with other works that use spectral methods or feature decorrelation techniques for different tasks or settings. It would be helpful to see how the proposed method relates to or differs from some related works.


**Reasons To Accept:**

1.	The paper provides an interesting perspective on the representation shift problem in CRE, and leverages spectral decomposition for avoiding representation shift. The authors propose a simple yet effective regularization technique that aims to increase the eigenvalues of the representation for each class and theoretically shows why feature decorrelation helps boost eigenvalues.
2.	The paper conducts experiments on two standard datasets and demonstrates the superior performance of the proposed method over existing baselines. The paper also conducts further analysis to verify the hypothesis that larger eigenvalues lead to better performance and vice versa.


**Reasons To Reject:**

1.	The paper does not provide much interpretation or intuition for the eigenvectors with larger eigenvalues, and what kind of features they capture. It would be preferable to provide some examples or visualizations from a high-level viewpoint.
2.	The paper does not compare or discuss the relation with other works that use spectral methods or feature decorrelation techniques for different tasks or settings. It would be helpful to see how the proposed method relates to or differs from some related works.


**Reproducibility:**

4: Could mostly reproduce the results, but there may be some variation because of sample variance or minor variations in their interpretation of the protocol or method.

**Reviewer Confidence:**

4: Quite sure. I tried to check the important points carefully. It's unlikely, though conceivable, that I missed something that should affect my ratings.

**Typos Grammar Style And Presentation Improvements:**

There are some typos and grammar mistakes, such as:
Abstract: eigenvalue > eigenvalues
Abstract : improve > improves

---

> ### Author Rebuttal · Authors · 2023-08-28
>
> Thank you for your comments and suggestions. Please find below our responses for your questions and concerns.
>
> **Reviewer**: *"The paper does not provide much interpretation or intuition for the eigenvectors with larger eigenvalues, and what kind of features they capture. It would be preferable to provide some examples or visualizations from a high-level viewpoint."*
>
> **Our Response**:
>
> Thank you for your suggestion. As discussed in our Limitations Section, we agree with the reviewer that we currently don't have clear interpretation for the the eigenvectors with larger eigenvalues. Based on your suggestion, we will further include visualizations for the eigendirections of the methods over our datasets to shed more light on their interpretation for continual relation extraction.
>
> **Reviewer**: *"The paper does not compare or discuss the relation with other works that use spectral methods or feature decorrelation techniques for different tasks or settings. It would be helpful to see how the proposed method relates to or differs from some related works."*
>
> **Our Response**:
>
> Thank you for your suggestion. In the paper, we have cited serval works on spectral decomposition [2,3] that is explored for self-supervised learning. The most related work to ours involves [1] that also studies spectral analysis for continual learning, aiming to boost the numbers and eigenvalues of the directions with larger eigenvalues to address catastrophic forgetting (called eigenvalue boosting). However, as presented in our response for reviewer Nh4p, our work is essentially different from [1] regarding the specific method to achieve eigenvalue boosting. In particular, while [1] employs a data augmentation method based on MixUp and data interpolation to implicitly boost eigenvalues, our work instead proposes a class-wise decorrelation feature regularization to explicitly promotes large eigenvalues and their numbers for continual learning. This difference brings important advantages for our method compared to that in [1], concerning better theoretical support for performance improvement (as shown in Appendix B.1), more efficient training resources and time (as we don't need to generate and consume augmented data), and better suite for discrete textual data in relation extraction. Our work is thus the first work to explore spectral analysis and feature regularization for continual relation extraction. We will carefully revise our paper to include these details and a more comprehensive section for related work.
>
> Also, thank you very much for your suggestions for our typos and grammar. We will revise our paper to fix these issues in the final version.
>
> [1] Zhu, Fei, et al., 2021. Class-incremental learning via dual augmentation. In NeurIPS 2022.
>
> [2] Adrien Bardes, Jean Ponce, and Yann LeCun. 2022. VICReg: Variance-invariance-covariance regularization for self-supervised learning. In ICLR 2022.
>
> [3] Tianyu Hua, Wenxiao Wang, Zihui Xue, Yue Wang, Sucheng Ren, and Hang Zhao. 2021. On feature decorrelation in self-supervised learning. In arXiv.

---

### Official Review · Reviewer_Nh4p · 2023-08-03

**Soundness:** 3

**Excitement:**

3: Ambivalent: It has merits (e.g., it reports state-of-the-art results, the idea is nice), but there are key weaknesses (e.g., it describes incremental work), and it can significantly benefit from another round of revision. However, I won't object to accepting it if my co-reviewers champion it.

**Paper Topic And Main Contributions:**

This paper focuses on Continual Relation Extraction (CRE), which involves continuous learning of new relations while maintaining proficiency in previously acquired relations. Inspired by existing continual learning approaches, the authors emphasize the issue of representation shift, a phenomenon where the learned deep space undergoes transformations during the learning process, leading to a decline in performance on earlier tasks. The authors present an intriguing perspective by exploring the representation shift phenomenon through a spectral viewpoint. They argue that the preservation of class shapes in the learned deep space is critical for mitigating performance degradation. They assume that when eigenvectors (or spectral components) associated with each class shape remain relatively unchanged, the shape is effectively preserved. To verify this assumption, the authors conduct a spectral experiment and demonstrate that eigenvectors with larger eigenvalues experience higher preservation after learning new tasks, indicating their efficacy in retaining class shapes. The paper then introduces a novel class-wise regularization approach aimed at enhancing eigenvalues during representation learning. This regularization method is shown to be effective in augmenting eigenvalues, which leads to better preservation of class shapes. Extensive experiments on two public benchmark datasets demonstrate performance gains in comparison to state-of-the-art models.

**Reasons To Accept:**

- The assumption (when eigenvectors (or spectral components) associated with each class shape remain relatively unchanged, the shape is effectively preserved) is technically sound and to some extent verified by the empirical experiments.
- Despite its simplicity, the proposed approach is able to achieve very promising results.

**Reasons To Reject:**

- The paper seems to be a trivial extension of [1] with application in CRE. The authors need to explain in detail the novelty of the paper and the differences from the existing works.
- The paper lacks sufficient ablation study to demonstrate the effectiveness and to verify the assumption of the relation between class shapes preservation and performance preservation.
- The empirical performance is not significantly better than baseline alternatives.

[1] Zhu, Fei, et al. "Class-incremental learning via dual augmentation." Advances in Neural Information Processing Systems 34 (2021): 14306-14318.

**Reproducibility:**

4: Could mostly reproduce the results, but there may be some variation because of sample variance or minor variations in their interpretation of the protocol or method.

**Reviewer Confidence:**

3: Pretty sure, but there's a chance I missed something. Although I have a good feel for this area in general, I did not carefully check the paper's details, e.g., the math, experimental design, or novelty.

---

> ### Author Rebuttal · Authors · 2023-08-28
>
> Thank you for your comments and suggestions. Please find below our responses for your questions and concerns.
>
> **Reviewer**: *"The paper seems to be a trivial extension of [1] with application in CRE. The authors need to explain in detail the novelty of the paper and the differences from the existing works."*
>
> **Our Response**:
>
> Thank you for your comment. Our work shares a similar perspective to [1] on spectral analysis for continual learning, aiming to boost the numbers and eigenvalues of the directions with larger eigenvalues to address catastrophic forgetting (eigenvalue boosting). However, we would like to emphasize a critical difference between our work and [1] (and existing related work) regarding the specific method to achieve eigenvalue boosting. In particular, [1] leverages a data augmentation method based on MixUp to randomly interpolate two images that can help artificially increase the number of classes for learning more transferable and diverse representations. This is thus an implicit approach for eigenvalue boosting. In contrast, our method introduces a class-wise decorrelation feature regularization that explicitly promotes large eigenvalues and their numbers for continual learning. In this way, our method enjoys better theoretical support (demonstrated in our Appendix B.1) for its effectiveness for continual learning, which cannot be achieved with the data augmentation method. In addition, the augmentation method in [1] in requires more expensive training computation and time than our method as it needs to generate the augmented data and learns from a larger set during the training process. Finally, it is important to note that the augmentation method in [1] is better suited for continuous data such images or speech, which can facilitate the interpolations between data samples. For relation extraction with discrete textual data as in our work, data interpolation will be more challenging, further highlighting the benefits of our regularization method to better support relation extraction. To our knowledge, we are also the first work to explore spectral analysis and feature regularization for continual relation extraction. We will clarify these details in our final version.
>
> **Reviewer**: *"The paper lacks sufficient ablation study to demonstrate the effectiveness and to verify the assumption of the relation between class shapes preservation and performance preservation."*
>
> **Our Response**:
>
> Thank you for your suggestion. In our experiments, we performed an ablation study to ablate the regularization component in our model that can be used to illustrate the relation between class shapes preservation and performance preservation, and we will include it in the final version. In particular, please see below the performance of our full model and ablated model for the regularization component over the TACRED dataset:
>
> | Model | T1 | T2 | T3 | T4 | T5 | T6 | T7 | T8 | T9 | T10 |
> | :-------------------------------- | :------: | :------: | :------: | :------: | :------: | :------: | :------: | :------: | :------: | :------: |
> | Our full model | 98.1 | 93.8 | 89.8 | 85.8 | 84.4 | 83.4 | 81.6 | 79.9 | 79.7 | 79.1 |
> | Our full model w/o regularization | 97.8 | 93.4 | 88.5 | 86.0 | 83.6 | 81.8 | 79.6 | 77.7 | 77.3 | 76.8 |
>
> As can be seen, the performance of the model drops significantly when the regularization component is excluded, testifying to its benefits for continual relation extraction. In addition, as shown in Equation (5) of Appendix B.1., our regularization component promotes the maintenance of uniform distribution of the eigenvalues across tasks, thus implicitly encouraging the class shape preservation for our continual model. As such, the removal of the regularization will likely translate into less preservation of class shapes, which also corresponds to worse performance for our model. In this way, our ablation study suggests certain connection between class shape preservation and performance preservation, and future work can further examine this issue to obtain deeper understanding.
>
> **Reviewer**: *"The empirical performance is not significantly better than baseline alternatives."*
>
> **Our Response**:
>
> Thank you for your comment. We would like to note that our performance improvement over baselines is similar and in the same range as those in many previous work with state-of-the-art performance and the same datasets for continual relation extraction, e.g., [2] and [3]. We thus believe our performance improvement can sufficiently demonstrate the effectiveness of the proposed method given current standards and progress in this area. We will clarify this information in our final version.
>
> [1] Zhu, Fei, et al. "Class-incremental learning via dual augmentation." Advances in Neural Information Processing Systems 34 (2021): 14306-14318.
>
> [2] Wang et al., 2022. Learning robust representations for continual relation extraction via adversarial class augmentation. In EMNLP 2022.
>
> [3] Zhao et al., 2022. Consistent Representation Learning for Continual Relation ExtractionIn. In Findings of ACL 2022.

---

### Official Review · Reviewer_oY2D · 2023-08-05

**Soundness:** 4

**Excitement:**

4: Strong: This paper deepens the understanding of some phenomenon or lowers the barriers to an existing research direction.

**Paper Topic And Main Contributions:**

This paper provides a novel spectral viewpoint on Continual Relation Extraction (CRE), in which a simple and effective class-wise feature decorrelation regularization is proposed to boost the eigenvalues of the representation for each class. Extensive experiments on two benchmark datasets demonstrate the effectiveness the proposed approach.

**Questions For The Authors:**

Overall, this study demonstrates a robust and well-designed approach. However, one aspect that could be addressed is the absence of a name for the proposed method.

**Reasons To Accept:**

1. The paper is well-written.
2. The spectral viewpoint is interesting and the proposed methodology is well-formulated.
3. Experiments validate that the proposed method outperforms compared baselines in most evaluation metrics.

**Reasons To Reject:**

1. To further strengthen this work, it would be valuable to include a comprehensive discussion on the comparison of computational costs between the proposed approach and existing state-of-the-art methods.

**Reproducibility:**

4: Could mostly reproduce the results, but there may be some variation because of sample variance or minor variations in their interpretation of the protocol or method.

**Reviewer Confidence:**

2: Willing to defend my evaluation, but it is fairly likely that I missed some details, didn't understand some central points, or can't be sure about the novelty of the work.

---

> ### Author Rebuttal · Authors · 2023-08-28
>
> Thank you for your comments and suggestions. Please find below our responses for your questions and concerns.
>
> **Reviewer**: *"To further strengthen this work, it would be valuable to include a comprehensive discussion on the comparison of computational costs between the proposed approach and existing state-of-the-art methods."*
>
> **Our Response**:
>
> Thank you for your suggestion. We agree with the reviewer that a comparison of computational costs will be helpful for further understand. In the table below, we report the training time of our models (with and without the feature regularization) and the previous state-of-the-art model CRL over two datasets TACRED and FewRel for 10 epochs.
>
> | Model | TACRED | FewRel |
> | :--------------------------------------- | :---------: | :---------: |
> | Our full model | 22 mins | 92 mins |
> | Our full model w/o regularization | 21 mins | 88 mins |
> |CRL (sota model) | 27 mins | 115 mins |
>
> As can be seen, CRL consumes much more time for training as it employs an expensive contrastive loss and updated memory bank. The feature regularization does increase the training time for our model a bit, but it is not very significant. We will include this comparison in our final version.
>
> **Reviewer**: *"Overall, this study demonstrates a robust and well-designed approach. However, one aspect that could be addressed is the absence of a name for the proposed method."*
>
> **Our Response**:
>
> Thank you for your suggestion. We will revise our paper and introduce a name for our method in the final version.

---

### Meta-Review · Area_Chair_JzxW · 2023-09-17

**Recommendation:** 4

**Metareview:**

Overall this paper makes a solid short research contribution, as also pointed out by the reviewers. The authors have provided sensible answers to the reviewers' questions, and these additional explanations should be incorporated in the paper to improve its readability.

Moreover, the point of motivating the Eigenvalue decomposition is also an important one.

---

### Decision · Program_Chairs · 2023-10-07

**Decision:**

Accept-Findings

**Comment:**

Overall this paper makes a solid short research contribution, as also pointed out by the reviewers. The authors have provided sensible answers to the reviewers' questions, and these additional explanations should be incorporated in the paper to improve its readability.

Moreover, the point of motivating the Eigenvalue decomposition is also an important one.